# Vitamin D upregulates the macrophage complement receptor immunoglobulin in innate immunity to microbial pathogens

Annabelle G. Small [1,2,3], Sarah Harvey[3], Jaspreet Kaur[3], Trishni Putty[2,3], Alex Quach [2,3], Usma Munawara[2,3,5], Khalida Perveen [1,2,3], Andrew McPhee[4], Charles S. Hii[2,3] & Antonio Ferrante [1,2,3 ✉]

Vitamin D deficiency remains a global concern. This 'sunshine' vitamin is converted through a multistep process to active 1,25-dihydroxyvitamin $D_3$ (1,25D), the final step of which can occur in macrophages. Here we demonstrate a role for vitamin D in innate immunity. The expression of the complement receptor immunoglobulin (CRIg), which plays an important role in innate immunity, is upregulated by 1,25D in human macrophages. Monocytes cultured in 1,25D differentiated into macrophages displaying increased CRIg mRNA, protein and cell surface expression but not in classical complement receptors, CR3 and CR4. This was associated with increases in phagocytosis of complement opsonised *Staphylococcus aureus* and *Candida albicans*. Treating macrophages with 1,25D for 24 h also increases CRIg expression. While treating macrophages with 25-hydroxyvitamin $D_3$ does not increase CRIg expression, added together with the toll like receptor 2 agonist, triacylated lipopeptide, Pam3CSK4, which promotes the conversion of 25-hydroxyvitamin $D_3$ to 1,25D, leads to an increase in CRIg expression and increases in CYP27B1 mRNA. These findings suggest that macrophages harbour a vitamin D-primed innate defence mechanism, involving CRIg.

[1] Department of Molecular and Cellular Biology, School of Biological Sciences, University of Adelaide, Adelaide, Australia. [2] The Robinson Research Institute and School of Medicine, University of Adelaide, Adelaide, Australia. [3] Department of Immunopathology, SA Pathology, Women's and Children's Hospital, Adelaide, Australia. [4] Department of Neonatal Medicine, Women's and Children's Hospital, Adelaide, South Australia. [5]Present address: Departement de Microbiologie et d'Infectiologie, Faculte de Medecine, Universite de Sherbrooke, Sherbrooke Quebec, Canada. ✉email: antonio.ferrante@adelaide.edu.au

Vitamin D is generated in humans by a two-step process. Firstly, the ultraviolet light band B (UVB) converts the cholesterol precursor 7-dehydrocholesterol to pre-vitamin D in the epidermis[1]. The second step involves the isomerisation to vitamin $D_3$ (or cholecalciferol) in a thermo-sensitive, non-catalytic reaction[1]. Vitamin $D_3$ is an inactive precursor which is bioactivated by the liver to form 25-hydroxyvitamin $D_3$ (25D). This is the main form of vitamin D present in the circulation and the form measured to determine 'vitamin D status' in an individual[2]. To form the biologically active metabolite, 1,25-dihydroxyvitamin $D_3$ (1,25D), 25D requires hydroxylation by the enzyme CYP27B1, or 25-hydroxyvitamin $D_3$ 1-α-hydroxylase. This is an intracellular process which occurs predominantly within the proximal and distal tubules of the kidneys but also extracellularly in activated macrophages[3,4]. 1,25D has been shown to play an important role in the killing of intracellular bacteria such as *Mycobacteria tuberculosis* and *M. leprae* in macrophages[5,6]. More recently, it has been shown that steroids and in particular dexamethasone increase the phagocytosis of microbial pathogens by macrophages[7–10]. In addition, in these studies, it was demonstrated that the steroids significantly increased the expression of the most recently described of the complement receptors, complement receptor immunoglobulin (CRIg) but not the classical complement receptors, CR3 and CR4. CRIg has been found to be a unique complement receptor which plays a key role in the phagocytosis and clearance of bacteria[11,12]. It was therefore of interest to see whether the steroid hormone properties of vitamin D regulated the expression of CRIg and phagocytosis in macrophages. We present evidence that shows that 1,25D promotes the dvelopment of human macrophages to express increased levels of CRIg at the mRNA, protein and cell surface expression which was associated with increased bacterial and fungal phagocytosis. The importance of innate immunity in promoting vitamin D effects was also demonstrated. While vitamin D, compared to 1,25D did not alter CRIg expression, addition of a TLR1/2 agonist in the presence of vitamin D led to increased expression of CRIg in association with increased expression of CYP27B1 which converts the 25D to 1,25D.

## Results and discussion

### 1,25D promotes the development of CRIg expressing macrophages.
Here we show that human macrophages differentiated from monocytes in the presence of 1,25D for 3 days, display increased CRIg mRNA expression (Fig. 1a, b). This effect is seen in a concentration-dependent manner over 0.5–200 nM (Fig. 1a, b). The increase induced by 1,25D on CRIg mRNA expression is seen in cultures initiated with either peripheral blood mononuclear cells (PBMC) (Fig. 1a) or purified monocytes (Fig. 1b). Because CRIg plays an important role in innate immunity[11,13,14], it was of interest to examine its expression in cord blood macrophages. CRIg expression was not significantly different between macrophages from adult and cord blood. Expression in cord blood macrophages was also upregulated by the presence of 1,25D (Fig. 1c). However, the data did trend towards a decrease in CRIg expression in cord blood macrophages, placing some reservation on this conclusion which should be resolved in further studies.

Further studies with purified monocytes show that the increase in CRIg expression is evident at the protein level and is reflected in an increase in the predominant isoform, the long (L) as well as the less prominent short (S) form, revealed by western blot analysis using a mouse anti-human CRIg monoclonal antibody (clone 3C9, Genentech, CA)[11] (Fig. 1d, Supplementary Fig. 1). Flow cytometry analyses of cell surface CRIg expression using the same monoclonal antibody show that macrophages derived from monocytes treated with 100 nM of 1,25D display significant increases in surface expression of

CRIg, compared with vehicle-treated control cells, suggesting that the increase in CRIg expression is likely to have an impact on cell function (Fig. 1d, Supplementary Fig. 1). Our data demonstrate that 1,25D also caused an increase in CRIg mRNA expression. This implies transcriptional regulation by the secosteroid. How this is mediated remains to be determined. Interestingly, neither the vitamin D receptor nor its heterodimeric binding partner, retinoid X receptor, both of which are required for CYP24 promoter activation[15], are amongst the 155 transcription factors that can bind the promoter/enhancer regions of the CRIg gene, *VSIG4*, as predicted by GeneHancer[16]. This raises the possibility that the transcriptional regulation of *VSIG4* by 1,25D does not involve a classical vitamin D response element such as the ones in the *CYP24A1* gene.

### Effects of 1,25D on expression of complement receptors 3 and 4.
As CRIg is not the only phagocytosis-promoting complement receptor expressed by macrophages[17], we next assessed the levels of the β-integrin complement receptors 3 and 4 (CR3 and CR4, respectively) in macrophages differentiated from monocytes in the presence of 1,25D, by measuring the levels of the α-subunits CD11b (CR3) and CD11c (CR4). There is no increase in CD11b mRNA. While there is a decrease in CD11c mRNA expression in these macrophages (Fig. 2a), this is not reflected in changes in either of these receptors at the protein level, revealed by western blot analysis (Fig. 2b, Supplementary Fig. 1), and in their cell surface expression, compared with untreated controls (Fig. 2c).

### 1,25D promotes macrophage phagocytosis.
With the finding that 1,25D upregulates CRIg, but not CR3 and CR4 in macrophages, we investigated whether the phagocytic capabilities of the cells were altered by the 1,25D treatment. Using commercially available *Staphylococcus aureus* bioparticles which fluoresce once within the phagosomes of the macrophage[18], we found that phagocytosis is significantly increased in 1,25D-treated cells, compared to untreated control cells (Fig. 2d). Using a second assay involving addition of heat-killed *Candida albicans* and analysis of cells under a microscope, phagocytosis is significantly higher in macrophages generated in the presence of 1,25D (Fig. 2e), with more particles engulfed per individual macrophage and more cells engulfing >4 particles. As the process of phagocytosis in both of these assays is promoted by complement and the other phagocytosis-promoting complement receptors CR3 and CR4 were essentially not influenced by 1,25D treatment, it can be tentatively concluded that the upregulation of phagocytic activity is most likely a direct result of the increase in CRIg expression on these cells. Interestingly, vitamin D or 1,25D has been associated with the promotion of M2 macrophage polarisation, a cell which is less inflammatory but has higher phagocytic activity than M1 macrophages[19–21]. In addition, CRIg is an important phagocytosis-promoting receptor able to mediate capture of bacterial, fungal, and parasitic pathogens[22], with increased phagocytic rates compared with CR3[11,12,23]. We have previously used the Santa Cruz monoclonal antibody to block CRIg function in dendritic cell-mediated T-cell response[24]. Attempts to address this issue with this blocking approach led to difficulties in interpretation of results. Blocking CRIg did not significantly decrease the phagocytosis of bacteria (Supplementary Fig. 2a). But an examination of CD11b expression demonstrated that the antibody caused a significant increase in this receptor (Supplementary Fig. 2b), most likely masking any depressed phagocytosis caused by the blocking of CRIg.

### 1,25D increases expression in mature macrophages.
Monocyte-derived macrophages have a lifespan ranging from weeks to years in the tissues[25]. As a result, these cells can potentially be exposed

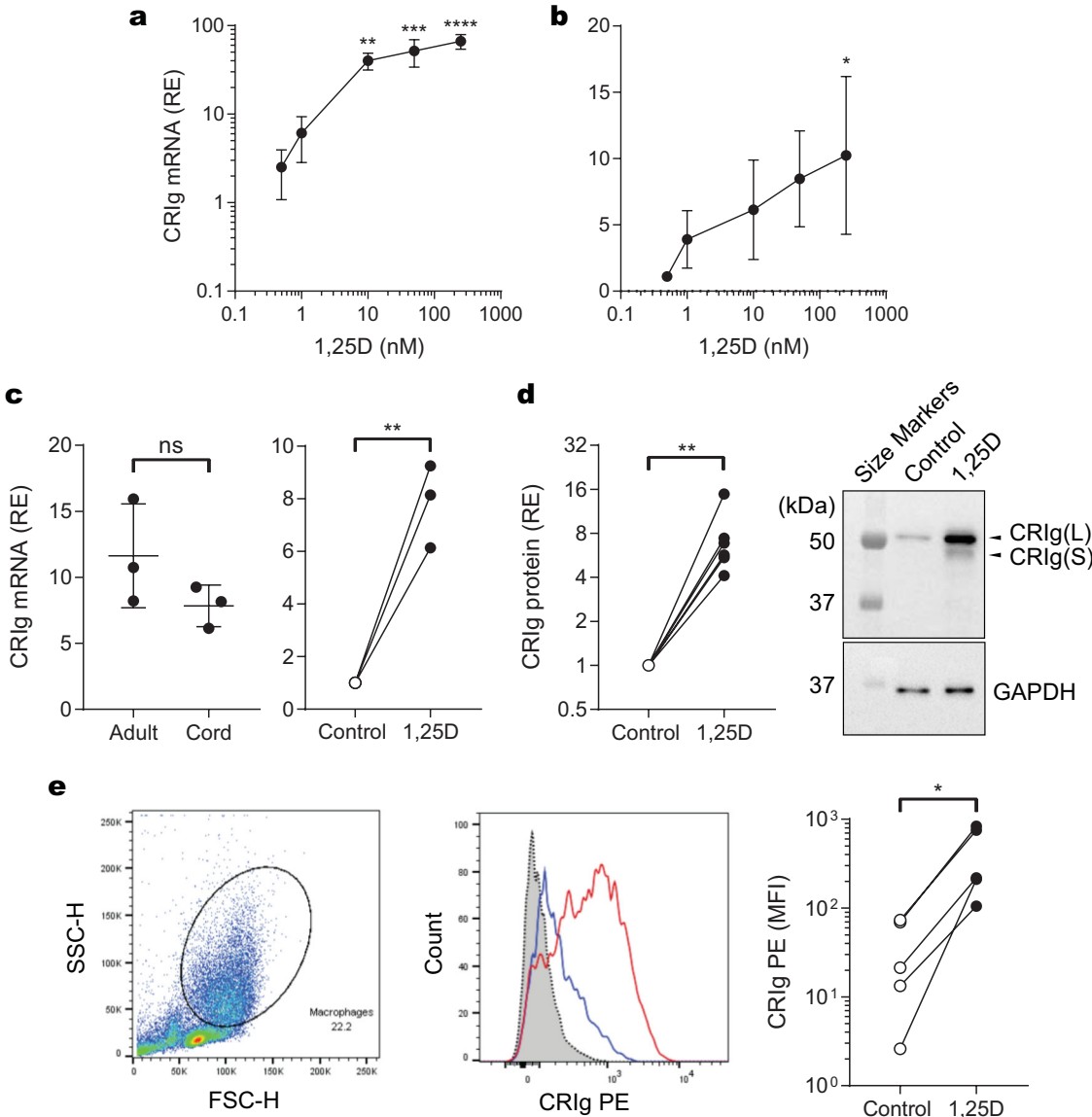

**Fig. 1 CRIg is upregulated in human macrophages by 1,25D.** PBMC or purified monocytes were cultured in the presence or absence of 1,25D. The cells were harvested to determine levels of CRIg mRNA on day 3 of culture, and CRIg protein on day 5 of culture. Relative expression (RE) of mRNA or protein was measured against GAPDH. **a** CRIg mRNA expression in PBMC cultured with varying concentration of 1,25D. **b** CRIg mRNA expression in macrophages derived from monocytes cultured with varying concentrations of 1,25D. **c** CRIg mRNA expression in macrophages derived from cord blood monocytes cultured for 3 days in the presence or absence of 100 nM 1,25D. **d** CRIg protein in macrophages derived from monocytes cultured in the presence or absence of 100 nM 1,25D. Western blot data are presented as fold-difference in CRIg band intensity normalized against GAPDH (loading control) with six experimental runs each with cells from a different individual. Representative western blot of CRIg expression (top panel) and GAPDH re-probe (bottom panel) are shown. **e** Macrophages derived from monocytes cultured in the presence or absence of 100 nM 1,25D were analysed for cell surface CRIg expression by flow cytometry. Left panel: Gating strategy based on size and granularity; centre panel: representative histogram overlay of CRIg expression: secondary antibody control is shown in dotted black, unstimulated macrophage CRIg fluorescence is shown in blue, and 1,25D stimulated macrophage CRIg fluorescence is shown in red; right panel: Δ median fluorescence intensity (MFI), for CRIg staining minus isotype control, is shown for control and 1,25D-treated cells from five individual experiments. **a, b** Data are presented as mean ± s.d. of three experiments each with cells from a different individual. $P$ values were calculated using one-way ANOVA followed by Dunnett's multiple comparison test. **c–e** Data are analysed by the paired, two-tailed Student's $t$-test. Statistical significance of 1,25D-treated versus controls are represented as *$P < 0.05$, **$P < 0.01$, ***$P < 0.001$, ****$P < 0.0001$.

to a range of homeostatic or inflammatory conditions. As their local microenvironment fluctuates, macrophages are able to display a high level of phenotypic plasticity reflecting this environment. Because of this, we sought to investigate whether adding 1,25D directly to the macrophages also causes a change in CRIg expression. Macrophages were prepared by incubating monocytes in culture for 5 days. These were then treated with 100 nM of 1,25D for 24 h. The macrophages show an increase in expression

of CRIg mRNA (Fig. 3a) and protein (Fig. 3b, Supplementary Fig. 1).

**Engagement of TLR1/2 on macrophages promotes pathways for generating 1,25D and increases in CRIg expression.** We surmise that macrophages with an active cytochrome P450 25-hydroxyvitamin D3-1alpha-hydroxylase (CYP27B1) and ability to

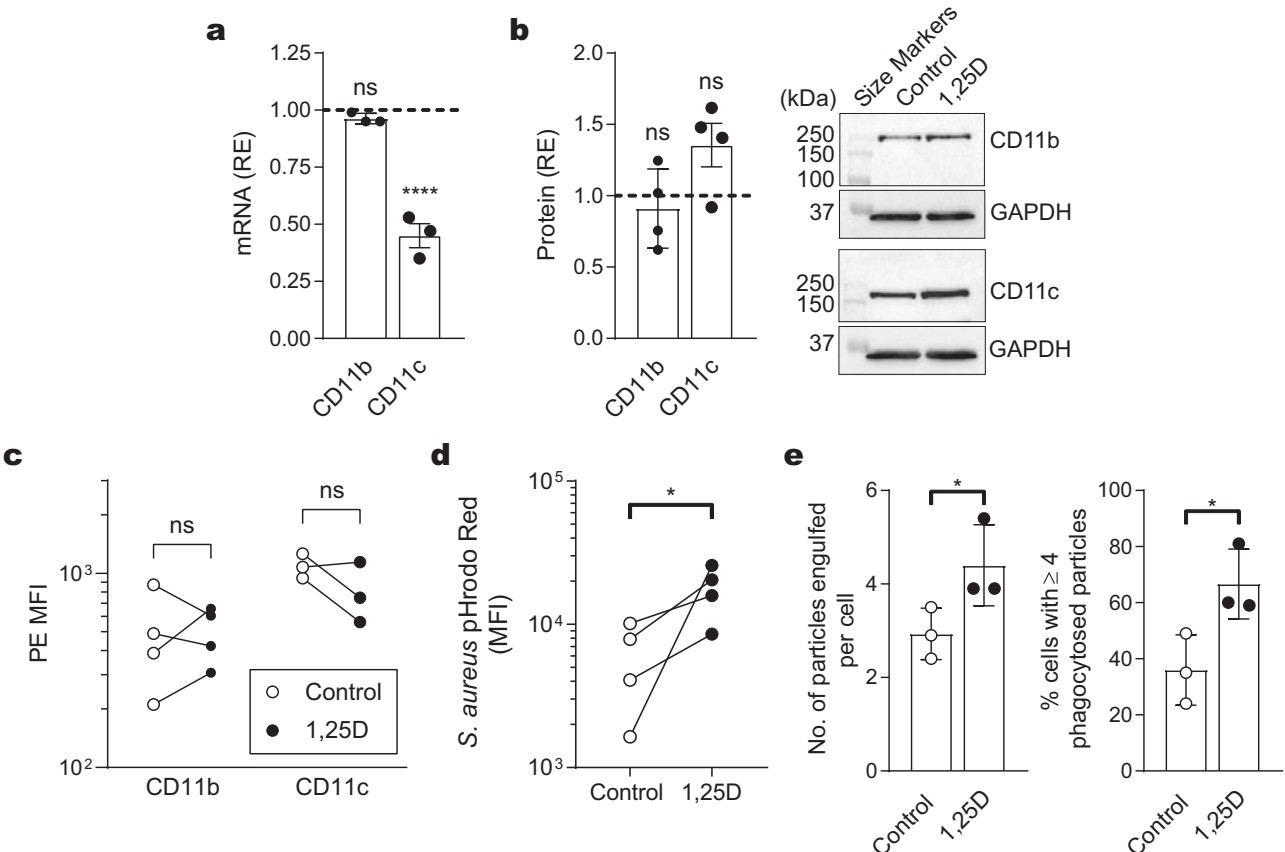

**Fig. 2 Effect of 1,25D on macrophage CR3 and CR4 expression and phagocytosis.** Monocytes were cultured for either 3 days (for mRNA expression) or 5 days (for protein expression) with 1,25D and examined for complement receptor expression. Relative expression (RE) of mRNA or protein was measured against GAPDH. **a** Macrophages were examined for CD11b and CD11c mRNA expression. Data are expressed as fold-change compared with untreated control from three experiments each conducted with cells from a different individual and expressed as mean ± s.d. **b** Macrophages were examined for CD11c and CD11b protein expression by western blotting, normalized against GAPDH from four experiments each with cells from a different individual. Representative western blots are shown. **c** Macrophages were analysed for CD11b and CD11c surface expression by flow cytometry. The PE MFI values are shown of four (CD11b) and three (CD11c) experiments, each conducted with cells from a different individual. **d** Phagocytosis of *S. aureus* bioparticles by macrophages as measured by the pH-sensitive pHrodo™ Red dye. Data are expressed as MFI, each conducted with cells from a different individual. **e** Phagocytosis of opsonized *C. albicans* by macrophages derived from monocytes cultured in either the presence or absence of 100 nM 1,25D for 5 days, is expressed as the number of engulfed particles per cell (left graph) and the percentage of cells with four or more phagocytosed particles (right graph). Data are presented as mean ± s.d. of three experiments each with cells from a different individual. **a, b** Data are presented as mean ± s.d. $P$ values were calculated using one-way ANOVA followed by Dunnett's multiple comparison test. **c–e** $P$ values were calculated using paired two-tailed (**c**) or one-tailed (**d, e**) Student's $t$-test. Significance of differences between 1,25D versus control, $*P < 0.05$, $****P < 0.0001$, ns = not significant.

convert the inactive 25D to 1,25D would show increased expression of CRIg, possibly through an autocrine or paracrine mechanism (Fig. 4a). The TLR1/2 agonist Pam3CSK4, is known to increase the expression of CYP27B1 in macrophages[26]. Using a combination of 25D and Pam3CSK4, we investigated whether treatment with these agents for 24 h causes an increase in CRIg expression. While treating macrophages with either 50 ng/mL Pam3CSK4 or 100 nM 25D independently has no significant effect, combined addition of these to cells causes an increase in CRIg mRNA and protein expression, particularly the long form (Fig. 4b, c, Supplementary Fig. 1). Furthermore, consistent with these findings was the result that treatment of macrophages with Pam3CSK4 caused a significant increase in their CYP27B1 mRNA expression (Fig. 4d). These results indicate that 1,25D produced by macrophages following engagement of TLR1/2[26], is able to act in an autocrine or intracrine manner to enhance CRIg expression.

Emerging interest in the non-classical biological effects of vitamin D has recently been highlighted[27], which includes an ability to regulate innate immune responses. Thus, 1,25D has been reported to increase the production of anti-microbial

peptides e.g. cathelicidin and β-defensin 2, and stimulate phagocytosis in macrophages[28]. Recently, the secosteroid has been shown to be required for IL-22 production by type 3 innate lymphoid cells and in defence against *Citrobacter rodentium* infection[29]. In macrophages, vitamin D is known to be required for defence against the intracellular pathogen *Mycobacterium tuberculosis*[3,30]. Macrophages express both the vitamin D receptor (VDR) and CYP27B1,[4] the latter enabling the generation of 1,25D[31]. VDR and CYP27B1 expression is upregulated by engaging TLR1/2 by triacylated lipoproteins on the microbial surface[3,32]. Another important piece of this immunobiology of the vitamin D 'jigsaw' puzzle shown by the present results is the upregulation of CRIg expression through the stimulation of TLR1/2 in the presence of 25D, providing evidence for a global role in anti-infective innate immunity. The results also make prominent the point that while CRIg is readily modulated, CR3 and CR4 are essentially not affected by 1,25D. It has been reported that cytokines and inflammatory mediators as well as the steroid drug dexamethasone display this differential effect on these receptors[7–9]. Our findings reveal an important mechanism

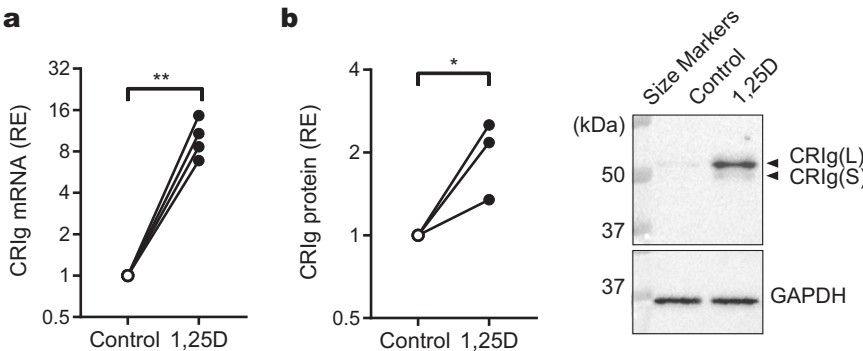

**Fig. 3 Effects of treating the macrophages directly with 1,25D on CRIg expression. a** Macrophages matured after 3 days of monocyte culture, were treated for a further 24 h with 100 nM of 1,25D or diluent and then the CRIg mRNA levels measured by qPCR. Data are expressed as CRIg relative to GAPDH from four experiments, each conducted with cells from a different individual. **b** Macrophages differentiated from culturing monocyte for 5 days culture, were treated as described above. The CRIg expression was measured by western blot in three experiments, each conducted with cells from different individuals. A representative western blot is shown of CRIg and GAPDH staining of the same blot. **a**, **b** Relative expression (RE) of mRNA or protein was measured against GAPDH. $P$ values were calculated by paired, one-tailed Student's $t$-test. Significance of differences between 1,25D versus control, *$P < 0.05$; **$P < 0.01$.

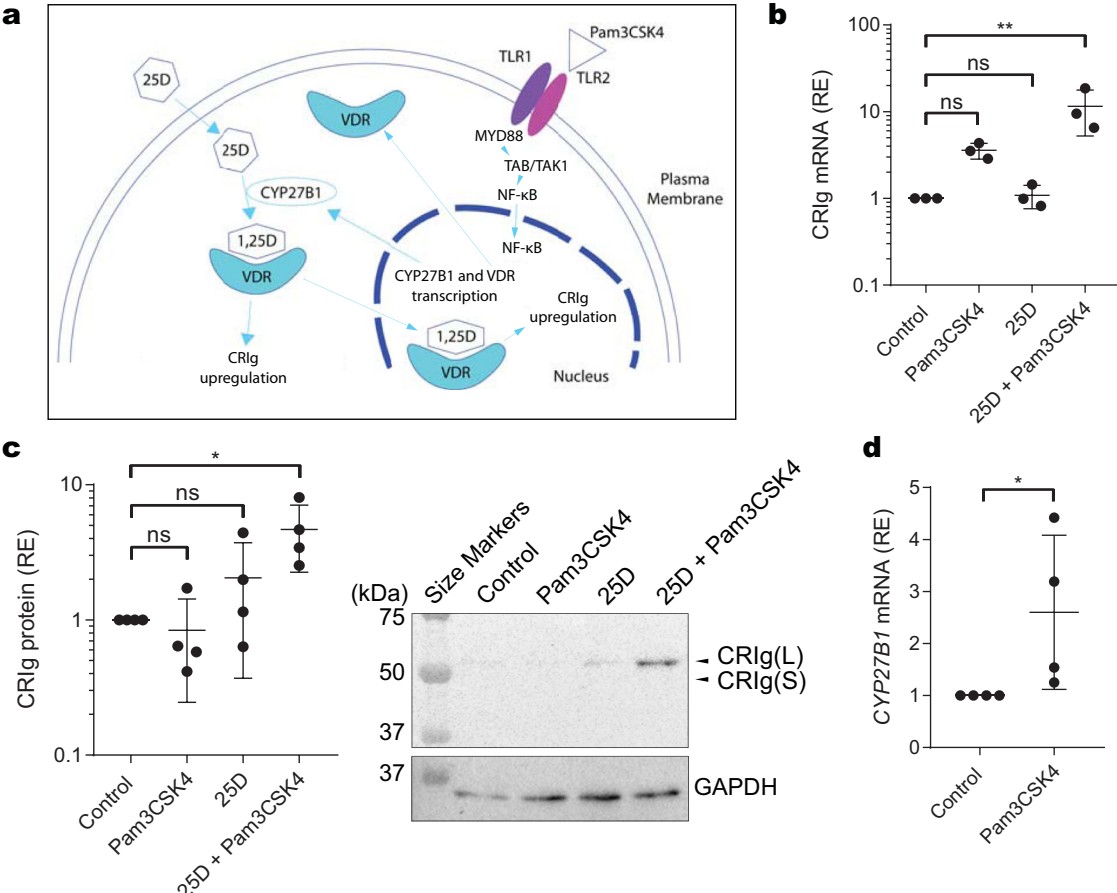

**Fig. 4 Vitamin D₃ promotes CRIg expression in macrophages treated with the TLR1/2 agonist Pam3CSK4. a** Schematic diagram showing engagement of TLR1/2 inducing enhanced expression of CYP27B1 which then converts 25D to 1,25D. **b** Macrophages matured after 3 days of monocyte culture, were treated for a further 24 h with either 50 ng/mL Pam3CSK4, 100 nM 25D or a combination of both or neither and the levels of CRIg mRNA determined. The levels were expressed relative to GAPDH mRNA (RE). Data are expressed as individual values and as means ± s.d. of three experiments. **c** Macrophages matured after 5 days of monocyte culture, were treated as described above. CRIg expression was measured by western blot relative to GAPDH expression. Data are expressed as means ± s.d. of five experiments together with a representative western blot. **d** For CYP27B1 expression, monocytes were differentiated to macrophages for 3 or 5 day, and Pam3CSK4 or control were added for 24 h and the levels of CYP27B1 mRNA determined by qRT-PCR. **b**, **c** $P$ values were calculated using one-way ANOVA followed by Dunnett's multiple comparison test. **d** $P$ value was calculated by the paired, one-tailed Student's $t$-test. Significance of differences between the different treatments are shown, *$P < 0.05$, **$P < 0.01$, ns = not significant.

in innate anti-microbial activity of macrophages, influenced by vitamin D. This study furthermore supports the importance of vitamin D sufficiency for a functional innate immune response, and supports the global concern of vitamin D deficiency[33].

## Methods

### Materials

*Human blood specimens.* The procurement of human blood and all experimental procedures were approved by the Human Research Ethics Committee of the Women's and Children's Health Network (WCHN), Adelaide, South Australia, in accordance with The National Statement on Ethical Conduct in Human Research (2007, updated 2018) (National Health and Medical Research Council Act 1992). Venous blood was collected from healthy adult volunteers by venipuncture with their informed consent, under approval number HREC/15/WCHN/21.

*Antibodies.* The mouse monoclonal antibody (clone 3C9, for flow cytometry, 0.2 μg; for western blotting, 1:3000) that recognizes the IgV domain of human CRIg was kindly provided by Dr. Menno van Lookeren Campagne (Genentech, San Francisco, CA). The rabbit recombinant monoclonal anti-CD11b antibody (ab133357, clone EPR1344, 1:1000), and mouse IgG1 isotype control antibody (ab37355) were purchased from Abcam. The mouse monoclonal anti-CD11c antibody (clone N-19, 1:1000) and goat PE-conjugated anti-mouse IgG antibody were purchased from Santa Cruz Biotechnology. The mouse monoclonal anti-GAPDH (clone 71.1, 1:20,000) was obtained from Sigma-Aldrich. The polyclonal HRP-conjugated rabbit anti-mouse (P0260), anti-goat (P0449) and goat anti-rabbit (P0448) immunoglobulin antibodies (1:2000) were obtained from Dako.

*Reagents.* Roswell Park Memorial Institute (RPMI) 1640 tissue culture medium, Hank's buffered saline solution (HBSS), foetal calf serum (FCS), L-glutamine, penicillin and streptomycin were purchased from SAFC Biosciences. Dithiothreitol (DTT), benzamidine, leupeptin, pepstatin A, phenylmethylsulfonyl fluoride (PMSF), 1α,25-dihydroxyvitamin D3 (1,25D) and 25-dihydroxyvitamin D3 (25D) were purchased from Sigma-Aldrich. Stock solutions of 1,25D and 25D were prepared to $10^{-3}$ M in 95% ethanol and stored at $-80\,°C$. Pam3CSK4 was purchased from Invivogen, with stock preparation at 1 mg ml$^{-1}$ in endotoxin-free water and storage at $-20\,°C$. Aprotinin was purchased from Merck.

### Cell preparation and culture

Peripheral blood mononuclear cells (PBMC) or cord blood mononuclear cells were prepared by density gradient centrifugation of blood on Ficoll-Paque PLUS (GE Healthcare). The interface layer containing PBMC was harvested and cells were washed in RPMI 1640 medium. Monocytes were purified from the MC following seeding of the latter at $2 \times 10^{7}$ per autologous plasma-coated 6-cm culture dish (TPP) and incubation at $37\,°C$, 5% $CO_2$/air, in a high humidity incubator for 2 h. Non-adherent cells were removed by three gentle washes resulting in >90% monocytes purity, and each dish replenished with 4 mL of RPMI 1640 supplemented with 2 mmol L$^{-1}$ L-glutamine, 100 U ml$^{-1}$ penicillin, 100 μg ml$^{-1}$ streptomycin and 10% FCS, pH 7.4. Experiments either utilized total PBMC or purified monocytes. Cells were stimulated with either, 1,25D, 25D, Pam3CSK4, or diluent and cultured for the duration specified in the 'Results' section. Cells were harvested after either 3 days (for CRIg mRNA analysis) or 5 days (for CRIg protein analysis or phagocytosis assays) culture by gentle scraping with a 'rubber policeman'.

### Quantitative PCR assays

The quantitative PCR (qPCR) assays were performed as previously described[8]. In brief, total RNA was extracted from harvested cells using TRIzol reagent (Invitrogen). cDNA was prepared using iScript cDNA synthesis kit (Bio-Rad). qPCR analysis was performed using iTaq™ Universal SYBR® Green Supermix (Bio-Rad) with the following conditions: initial denaturation for 5 min at $95\,°C$ followed by 40 cycles at $95\,°C$ for 30 s, $60\,°C$ for 30 s and $72\,°C$ for 30 s using an iQ5 Real-Time Detection System with iQ5 Optical System v2.1 software (Bio-Rad). Data were normalized to the expression of a control gene GAPDH for each experiment. The primer pairs used were for human CRIg (Forward: 5′-ACACTT ATGGCCGTCCCAT-3′; Reverse: 5′-TGTACCAGCCACTTCACCAA-3′), CD11b (F: 5′-CCTGGTGTTCTTGGTGCCC-3′ and R: 5′-TCCTTGGTGTGGCACGTAC TC-3′) CD11c (F: 5′-CCGATTGTTCCATGCCTCAT-3′; R: 5′-AACCCCAATTGC ATAGCGG-3′), CYP27B1 (F: 5′-TGGCCCAGATCCTAACACATTT-3′) (R: 5′-G TCCGGGTCTTGGGTCTAACT-3′)[34] and GAPDH (F: 5′-GAGTCAACGGATTT GGTCGT-3′; R: 5′-GACAAGCTTCCCGTTCTCAGCCT-3′).

### Phagocytosis assays

*Staphylococcus aureus bioparticle uptake quantitation by flow cytometry.* Briefly, $1 \times 10^{6}$ macrophages in HBSS with 8% human AB serum, were incubated with 80 μg pHrodo™ Red *S. aureus* Bioparticles™ (Invitrogen), in a final volume of 400 μL in $12 \times 75$ mm round-bottom tubes[18]. These were gassed with 5% $CO_2$/air and capped, with incubation at $37\,°C$ for 1 h. Following washing in HBSS, samples were acquired using a BD FACSCanto I flow cytometer with FACSDiva 8.0, with

analysis using FlowJo 10.1 software (FlowJo LLC) to determine bioparticle uptake by changes in median fluorescence intensity in the PE channel. For experiments run in the presence of the anti-CRIg blocking antibody, $1 \times 10^{6}$ macrophages were incubated at room temperature with either 10 μg/mL mouse anti-human CRIg monoclonal antibodies (clone 6H8, Santa Cruz Biotechnology; clone 6C9, Genentech) or mouse IgG1 isotype control antibodies for 15 min prior to conducting the phagocytosis assay as above.

*Candida albicans particle uptake quantitation by microscopy.* This phagocytosis assay was performed essentially as described previously[7,8]. Briefly, $1 \times 10^{5}$ *C. albicans* yeast particles were added to $5 \times 10^{4}$ macrophages in a final volume of 0.5 ml HBSS. Complement-containing human AB serum was added to a final concentration of 10%. The cells were incubated for 15 min at $37\,°C$ on a rocking platform. Following removal of unphagocytosed yeast particles by differential centrifugation at $175 \times g$ for 5 min, the remaining macrophages in the pellet were cytocentrifuged onto a microscope slide and stained with Giemsa. The particles in phagocytic vacuoles were enumerated, with phagocytosis was scored as both the number of macrophages that had engulfed >4 fungi as well as the number of fungi engulfed per cell.

### Assessing cell surface CRIg and CD11b expression

Macrophage surface CRIg expression was determined by flow cytometry[7]. Briefly, harvested cells were incubated in $12 \times 75$ mm round-bottom tubes on ice with 100 μg purified human IgG (Kiovig, Baxter) for 15 min. This was followed by addition of 0.2 μg of either anti-human CRIg or mouse IgG1 isotype control antibodies, with further incubation for 20 min. Cells were washed with 2 mL PBS with centrifugation at $500 \times g$ for 5 min. Goat anti-mouse IgG PE secondary antibody was then added, with continued incubation in the dark on ice for 20 min. For experiments assessing CD11b, 0.2 μg of anti-CD11b (PE) were added to concurrent tubes set up in the absence of anti-CRIg primary antibody. Following washing twice more, the cells were acquired (50,000 event minimum) on a BD FACSCanto I with FACSDiva 8.0, and data analysed using FlowJo 10.1.

### Western blotting assays

Protein analysis in harvested macrophages was performed using western blot essentially as previously described[8]. Lysates were generated from macrophages in each culture dish with 100 μL of buffer containing 20 mmol L$^{-1}$ HEPES, pH 7.4, 0.5% Nonidet P-40 (v/v), 100 mmol L$^{-1}$ NaCl, 1 mmol L$^{-1}$ EDTA, 2 mmol L$^{-1}$ $Na_3VO_4$, 2 mmol L$^{-1}$ DTT, 1 mmol L$^{-1}$ PMSF and 1 μg mL$^{-1}$ of each protease inhibitor, benzamidine, leupeptin and pepstatin A. Total protein in the soluble fractions was quantitated using the Qubit™ Protein Assay Kit on a Qubit 3.0 (Invitrogen), prior to the addition of Laemmli buffer. Samples were boiled at $100\,°C$ for 5 min and 60 μg of protein were subjected to 10% SDS-PAGE at 170 V for ~1 h, using the Mini-PROTEAN 3 system (Bio-Rad). The samples were transferred onto nitrocellulose membrane using the Trans-Blot® Turbo™ Transfer System (Bio-Rad). The extent of protein transfer was ascertained using 0.1% Ponceau S membrane staining. After blocking in TBST with 5% skim milk (blocking solution), the membrane was incubated with either mouse anti-human CRIg, rabbit anti-human CD11b, or mouse anti-human CD11c antibodies in blocking solution overnight at $4\,°C$. The membrane was washed in blocking solution ($3 \times 5$ min) and then incubated with the appropriate secondary HRP-conjugated antibody (anti-mouse, anti-rabbit or anti-goat IgG) in blocking solution for 1 h at room temperature. Immunoreactive material was detected using the Western Lightning Plus-ECL Enhanced Chemiluminescence Substrate (Perkin-Elmer), with protein bands visualized on a ChemiDoc™ XRS+ Imager and quantitated using Image Lab™ Software, Version 3.0 (Bio-Rad). For GAPDH determination, stained membranes were subjected to antibody stripping using ReBlot Plus Mild Solution (Millipore) and incubated with mouse anti-human GAPDH antibody, followed by the staining and visualization steps as described above.

### Statistics and reproducibility

Graphpad Prism 8.0 (Graphpad Software) was used for statistical analysis. Mean differences were compared using *t*-tests (for comparisons of two groups) or one-way ANOVA followed by multiple comparison tests (for comparisons of three or more groups). *P* values <0.05 were considered to be statistically significant.

**Reporting summary.** Further information on research design is available in the Nature Research Reporting Summary linked to this article.

## Data availability

The data supporting this study are available within the paper and Supplementary Information. Source data can be found in Supplementary Data 1. Any additional data relating to the study are available from the corresponding author on reasonable request.

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

## Acknowledgements

We are grateful to Dr Menno van Lookeren Campagne, Genentech for providing us with the mouse anti-human CRIg monoclonal antibody, clone 3C9. The work received funding from the Women's and Children's Hospital Network Research Foundation of South Australia and the National Health and Medical Research Council of Australia. A. G.S. was a recipient of an Australian Government Research Training Programme Scholarship.

## Author contributions

A.G.S., S.H. and A.F. designed the experiments. A.G.S., S.H., T.P., J.P., U.M. and K.P. carried out the experiments. A.Mc.P. assisted with the cord blood cell study and critical reading of the manuscript. A.G.S., A.Q., C.S.H., and A.F. were involved in collating data, statistical analyses, data interpretation and writing of the manuscript. A.F. initiated and supervised the project, and A.G.S. and A.F. were responsible for drafting the manuscript.

## Competing interests

The authors declare no competing interests.
