## [Peer Review File · Communications Biology]

Reviewers' comments:

Reviewer #1 (Remarks to the Author):

The manuscript by Small and colleagues reports on some novel effects of 1,25D to increase complement receptor immunoglobulin on human macrophage. They do a sound job of demonstrating mRNA and protein upregulation with 1,25D that corresponds to an increase in phagocytosis capacity. The data does a good job of demonstrating an additional mechanism that underlies the anti-infectious effects of vitamin D. However a couple of overinterpretations of the data undermine several of the other conclusions.

Specifically:

1) Figure 4 does not provide evidence of Cyp27B1 activity in the macrophage. 25D is not inactive it is less active than 1,25D and if there is no 1,25D than 25D can bind to the vitamin D receptor. The kd for 1,25D binding to the VDR is in the nM range and 25 is about 100 fold higher. A dose of 100nM 25D could very well be binding to the VDR without being converted to 1,25D. The data showing the increase in complement receptor immunoglobulin with activation and 25D is not convincing. Without any measures of 1,25D production or other evidence of induction of Cyp27B1 the data provided in Figure 4 cannot be interpreted to mean that autocrine production of 1,25D leads to upregulation of complement receptor immunoglobulin.

2) Figure 4A suggests nongenomic and an illy defined mechanism for the increase in complement receptor immunoglobulin. There is no evidence provided here for nongenomic effects of vitamin D. Instead the data show that the increase in protein goes along with an increase in mRNA for complement receptor immunoglobulin. Based on the comment (bottom p. 3) that there is no vitamin D response element in this gene the authors discount transcriptional regulation. It has become clear that the vitamin D regulated transcription of genes often does not include a classical vitamin D response element such as the ones in the Cyp24A1 gene. Without further data to support a different mechanism transcriptional regulation cannot be discounted.

Reviewer #3 (Remarks to the Author):

This study addresses another mechanism by which the active form of vitamin D regulates innate immunity in human macrophages - the induction of expression of CRIG, a complement receptor for C3 proteins. Apart from minor concerns (below), the data presented are quite convincing. However, as is stands, the study is descriptive and correlative in nature; in other words enhanced CRIG expression correlates with enhanced phagocytosis, but its contribution to phagocytosis has not been addressed. The authors should perform experiments with available CRIG blocking antibodies or peptides to provide evidence that it is indeed elevated CRIG expression that contributes to enhanced phagocytosis in 1,25D-treated cells.

Minor comments.

1. There is no legend to Fig. 1b.
2. It is unfortunate that the adult and cord blood samples in Fig. 1C were only obtained in triplicate. More replicates would be required to confirm that there is indeed no significant difference in CRIG expression between the two sources. Conclusions in the manuscript should be tempered to reflect this.
3. Line 74: "since the presence of a vitamin D receptor (VDR) binding site has not been 74 predicted in the promoter regions of the VSIG4 gene". Please reference or describe the sources of information used to arrive at this conclusion.
4. Reference 19 does not seem to focus on the "the non-classical biological effects" but rather non-genomic actions. The following would be more appropriate: <https://doi.org/10.1210/er.2018-00126>.

Response to Reviewers comments

We are grateful for the comments and suggestions made by both reviewers. We essentially agree with the comments made and have added additional data and revised the manuscript accordingly. The changes have been highlighted in yellow background. Thank you.

Response to Reviewer 1

Reviewer #1 (Remarks to the Author):

The manuscript by Small and colleagues reports on some novel effects of 1,25D to increase complement receptor immunoglobulin on human macrophage. They do a sound job of demonstrating mRNA and protein upregulation with 1,25D that corresponds to an increase in phagocytosis capacity. The data does a good job of demonstrating an additional mechanism that underlies the anti-infectious effects of vitamin D. However, a couple of over interpretations of the data undermine several of the other conclusions.

Specifically:

1) Figure 4 does not provide evidence of Cyp27B1 activity in the macrophage. 25D is not inactive it is less active than 1,25D and if there is no 1,25D than 25D can bind to the vitamin D receptor. The K_d for 1,25D binding to the VDR is in the nM range and 25 is about 100-fold higher. A dose of 100nM 25D could very well be binding to the VDR without being converted to 1,25D. The data showing the increase in complement receptor immunoglobulin with activation and 25D is not convincing. Without any measures of 1,25D production or other evidence of induction of Cyp27B1 the data provided in Figure 4 cannot be interpreted to mean that autocrine production of 1,25D leads to upregulation of complement receptor immunoglobulin.

Response. Thank you for this very important point. In our initial attempts to look at the 1,25D levels in supernatants and cell lysates we have asked our colleagues at SA Pathology services to measure these for us as it requires special instruments. However, despite their efforts they were not able to measure this accurately using the DiaSorin XL 1,25 Dihydroxyvitamin D immunoassay kit (product code 310980) as there seems to be some interference by the presence of 25D *per se*. When we re-consulted the literature where Pam3CSK4 was used to establish the concept of TLR2 induced up regulation of CYP27B1, they had measured CYP27B1 mRNA. So, we have followed the suggestion of the reviewer to measure CYP27B1 mRNA levels. The additional data presented in Fig. 4d show that Pam3CSK4 treatment leads to a significant increase in CYP27B1 mRNA. This has now been added to Abstract, page 2, line 16-17, also added to page 6, line 4-6, and methods page 12, line 20-21.

2) Figure 4A suggests nongenomic and an illy defined mechanism for the increase in complement receptor immunoglobulin. There is no evidence provided here for nongenomic effects of vitamin D. Instead the data show that the increase in protein goes along with an increase in mRNA for complement receptor immunoglobulin. Based on the comment (bottom p. 3) that there is no vitamin D response element in this gene the authors discount transcriptional regulation. It has become clear that the vitamin D regulated transcription of genes often does not include a classical vitamin D response element such as the ones in the Cyp24A1 gene. Without further data to support a different mechanism transcriptional regulation cannot be discounted.

Response. Thank you and we agree that our statement needs to be refined as pointed out by the reviewer. We have modified this statement which now appears on page 3 line 24 to page

4 line 6. The diagram (Fig. 4a) has also been modified accordingly. Note that two additional references have been added, ref 10 and 11.

Reviewer #3 (Remarks to the Author):

This study addresses another mechanism by which the active form of vitamin D regulates innate immunity in human macrophages - the induction of expression of CRIg, a complement receptor for C3 proteins. Apart from minor concerns (below), the data presented are quite convincing. However, as is stands, the study is descriptive and correlative in nature; in other words enhanced CRIg expression correlates with enhanced phagocytosis, but its contribution to phagocytosis has not been addressed. The authors should perform experiments with available CRIg blocking antibodies or peptides to provide evidence that it is indeed elevated CRIg expression that contributes to enhanced phagocytosis in 1,25D-treated cells.

Response. Thank you and we agree that this would enhance the understanding of mechanisms. The difficulty with using blocking peptides is that they may also block the engagement of complement opsonised bacteria to CR3. We have managed to block CRIg function in DCs (ref 20) using anti-CRIg monoclonal antibody. We thus used this approach. However, despite many efforts we could not block this increased phagocytosis using the blocking antibody. We then surmised that the antibody binding to neutrophils may cause cell activation and an increase in CR3 expression thereby masking any decrease ensuing from the blocking. Examination of CR3 in anti-CRIg antibody treated neutrophils showed a significant increase in expression of CR3. We have explained the limitation raised by the reviewer in the revised manuscript and also why this point is difficult to prove. We have added the word 'tentatively' before concluded on page 4, last line. There is an explanation to our attempt to block CRIg on page 5 line 6-12. The data has been added as extended data Fig 2. Other changes appear in Methods page 12 and 13.

Minor comments.

1. There is no legend to Fig. 1b.

Response. Fig. 1b does contain a legend, see line 5.

2. It is unfortunate that the adult and cord blood samples in Fig. 1C were only obtained in triplicate. More replicates would be required to confirm that there is indeed no significant difference in CRIg expression between the two sources. Conclusions in the manuscript should be tempered to reflect this.

Response. We agree that there is a trend for lower levels in cord blood macrophages, despite this not being significant. We have modified the statement in the text page 3 line 11 to reflect this as suggested by the reviewer. Thank you.

3. Line 74: "since the presence of a vitamin D receptor (VDR) binding site has not been predicted in the promoter regions of the VSIG4 gene". Please reference or describe the sources of information used to arrive at this conclusion.

Response. Thank you. This has now been described on page 3 line 24 to page 4 line 6, and other information /sources has been added, ref 10 and 11.

4. Reference 19 does not seem to focus on the “the non-classical biological effects” but rather non-genomic actions. The following would be more appropriate: <https://doi.org/10.1210/er.2018-00126>.

Response. Yes agree, the reference has been changed as suggested by reviewer, now ref 22.
Thank you.

REVIEWERS' COMMENTS:

Reviewer #1 (Remarks to the Author):

Acceptable as revised.

Reviewer #3 (Remarks to the Author):

The authors have appropriately addressed reviewer concerns.